# Effect of Resin Bleed Out on Compaction Behavior of the Fiber Tow Gap Region during Automated Fiber Placement Manufacturing

**DOI:** 10.3390/polym16010031

**Published:** 2023-12-21

**Authors:** Von Clyde Jamora, Virginia Rauch, Sergii G. Kravchenko, Oleksandr G. Kravchenko

**Affiliations:** 1Department of Aerospace and Mechanical Engineering, Old Dominion University, Norfolk, VA 23529, USA; vrauch.vmr@gmail.com (V.R.); okravche@odu.edu (O.G.K.); 2Department of Materials Engineering, The University of British Columbia, Vancouver, BC V6T 1Z4, Canada; sergey.kravchenko@ubc.ca

**Keywords:** process modeling, compaction, finite element analysis, automated fiber placement

## Abstract

Automated fiber placement is a state-of-the-art manufacturing method which allows for precise control over layup design. However, AFP results in irregular morphology due to fiber tow deposition induced features such as tow gaps and overlaps. Factors such as the squeeze flow and resin bleed out, combined with large non-linear deformation, lead to morphological variability. To understand these complex interacting phenomena, a coupled multiphysics finite element framework was developed to simulate the compaction behavior around fiber tow gap regions, which consists of coupled chemo-rheological and flow-compaction analysis. The compaction analysis incorporated a visco-hyperelastic constitutive model with anisotropic tensorial prepreg viscosity, which depends on the resin degree of cure and local fiber orientation and volume fraction. The proposed methodology was validated using the compaction of unidirectional tows and layup with a fiber tow gap. The proposed approach considered the effect of resin bleed out into the gap region, leading to the formation of a resin-rich pocket with a complex non-uniform morphology.

## 1. Introduction

High-throughput manufacturing processes for fiber-reinforced composites allow for complex layups with a lower processing cycle time [1,2]. Particularly, automated fiber placement (AFP) is a robotic system that deposits pre-impregnated (prepreg) fiber tow strips at specified locations and enables tailored orientations [3,4,5,6,7]. For example, Wu et al. created a variable stiffness structure with a cut-and-restart functionality of an AFP machine in order to manufacture a prototype of an airplane fuselage [3]. However, deposition induced AFP tow features, especially fiber tow gaps, inevitably lead to irregular, local non-uniform morphology upon cure and compaction [8]. Manufacturing features, such as voids and resin-rich regions, develop when plies neighboring to the gap tow sink into the gap and experience large non-linear deformation caused by the flow of liquid resin into the empty space (Figure 1). Previous reports show that AFP manufacturing defects created during the early stages of cure before resin gelation can significantly affect the mechanical properties of the composites [8,9,10]. These regions with irregular morphology can serve as weak points and reduce the strength [11,12,13].

Fiber tow gaps and overlaps can be restricted during manufacturing or allowed within a specific tolerance. However, imperfections in the cured composites propagate from these AFP deposition features within the preform and produce local non-uniform morphology, such as ply thickness variations, waviness, and resin-rich pockets (Figure 1). These irregularities produced from AFP deposition features cannot be entirely avoided and should be considered as a part of AFP-manufactured composites to properly predict the strength of composite structures [14,15]. Furthermore, microscopic features, such as fiber volume fraction redistribution, and mesoscopic features (ply thickness, waviness) are not formed in isolation from each other and should be analyzed within a comprehensive framework that considers their coupling [7,16,17].

Prior to vitrification, resin changes from an incompressible viscoelastic liquid to a viscoelastic solid upon gelation. The compaction of AFP-deposited preform develops prior to composite gelation, wherein resin viscosity depends on the degree of cure and temperature [18,19]. At this stage, the deformation of the prepreg acts as a homogeneous medium, which is treated as a squeeze flow. In the case of AFP preform compaction, the squeeze flow develops when the fiber tow above the gap region is pushed inside of the void space during the compaction. However, resin bleed out through the fiber bed also occurs simultaneously (Figure 2a), causing the formation of resin-rich zones in the tow gap void and local fiber volume fraction variation in the composite [20]. Conventionally, the effect of resin bleed out in manufacturing of prepregs is considered insignificant, but in the case of AFP preforms with fiber tow gaps, it was shown to affect the waviness of the ply [21].

Recently, a hyper-viscoelastic material model was applied by Belnoue et al. to simulate the squeeze flow deformation during compaction [7,22,23,24,25]. However, the research did not consider the anisotropic viscosity of prepreg tape and its dependence on the fiber volume fraction. Furthermore, earlier studies did not consider the effects of the resin bleed out from the compacted prepreg into the gap region. The present study considers the effect of resin bleed out on the formation of non-uniform morphology around the region of the fiber tow gap. To model prepreg behavior around the internal void, like in the case of debulking, a negative internal pressure gradient can be considered inside of the void using porous-cohesive elements [26]. Another approach is to consider the flow front filling the gap via a one-dimensional numerical model [27]. These studies represent a two-phase modeling scheme where a region other than the composite is represented [21,27]. In the present work, a single-phase model was used and the effects of the resin bleed out into the gap was shown to have a larger impact on the morphology of the composite prior to resin gelation.

The present treatise extends the use of a hyper-viscoelastic model by introducing the fiber volume-dependent anisotropic tensorial viscosity of the prepreg into the formulation of the viscous component [22,28,29]. This approach introduces the local fiber volume fraction as a coupling between the squeeze flow and resin bleed out. Resin bleed-out measurements were obtained from the unidirectional tape compaction experiments and used to predict the amount of resin bleed out around the tow gap region. In the present case, a single-phase model was developed based on sequential compaction flow analysis, which adds the influence of resin flow into fiber tow gaps without explicitly modeling the resin. First, the hyper viscoelastic finite element analysis (FEA) squeeze flow was performed and the tow gap volume decrease was predicted. Once the tow gap volume reached the resin bleed-out volume, the time of filling was predicted. To capture the presence of resin inside of the tow gap, internal pressure was introduced in the tow gap based on the predicted time of void filling. The interaction between the squeeze flow and resin bleed out was investigated and the predicted resin-rich pocket shape agreed closely with the experimental observations. The proposed sequential compaction flow methodology can be extended to modeling various combinations of fiber tow gaps in AFP manufacturing to predict fiber waviness, resin-rich regions, as well as ply thickness variation in the region around the fiber tow gaps.

## 2. Sample Fabrication and Experimental Measurement

### 2.1. Unidirectional Tow Compaction Experiments

Experiments were performed on unidirectional samples with different geometries designed to measure the tow spreading and to predict resin bleed out from the prepreg during compaction. The material used was tows of IM7/8552 (Hexcel, Stamford, CT, USA) with low tack behavior with a width of 6.57 mm and a thickness of 0.14 mm. To measure the compaction of unidirectional tows in the early stages of manufacturing before resin gelation, the first hold stage of the manufacturer’s cure cycle was used. This included a hold temperature of 110 °C with that of an initial ramp of 3 °C/min. A pressure of 0.551 MPa (80 psi) was applied to the composite at a ramp of 0.034 MPa/min. Six configurations were made for the unidirectional samples. The tow width and length were varied between one, two, and four plies. This measured to be 0.14 mm, 0.28 mm, and 0.56 mm for the thickness and the widths were 6.46 mm, 12.92 mm, and 25.84 mm, respectively. The length of the samples was a constant 25.84 mm. Six samples were made for each of the configurations and the experimentally measured results were averaged for each configuration.

To analyze the tow spreading during cure, the thickness and width of the unidirectional samples were measured at 5, 15, 30, and 60 min into the hold stage by taking images of the top *x*-*y* plane of the unidirectional samples using a digital microscope. At the end of the first isothermal stage of the cure cycle, the bleed-out resin was observed on the outside of the ply (Figure 2b). This resin was cut with a blade and its mass was measured. The images taken throughout the cure cycles were used to estimate the resin bleed out in fiber as well as transverse to the fiber direction throughout the cure process.

### 2.2. Compaction Experiments with an Embedded Fiber Tow Gap

A cross-ply configuration with an embedded tow gap was laid up and used to study the compaction behavior around the region of the tow gap. The fiber tow stacking sequence [0°/90°/0°] was used with the centrally located 90° tow gap oriented in the *y* direction. The width of the sample consisted of four fiber tows with 0° and two tows with 90° layers (Figure 3). The overall dimensions of the fabricated sample before cure were 25.15 mm × 25.15 mm × 0.40 mm. A hand roller was used to apply slight uniform contact force to ensure proper contact in the layup. Once the top 0° layers were deposited, no additional fiber spreading was observed and the 0° tows sank into the tow gap region in the central location of the gap.

The same cure cycle was used to cure the tow gap sample, which was post-cured in the oven at 180 °C for two hours. To observe the cross section of the fiber tow gap, the samples were cut and polished. To determine the effects of the compaction on the cured morphology, the *x*-*z* plane of the samples was analyzed using a Leica DM6 upright microscope, and different ply features were measured, including the fiber angles in the top 0° tow, size, and shape of the resin-rich region, as well as ply thickness variations in the bottom 0° tow.

## 3. Multiphysics FEA Simulation Framework for AFP Fiber Tow Preform Compaction

The interconnected nature of the cure and compaction process was studied via a unified process modeling approach in the form of a physics-based simulation that captured the chemo-rheological transformation of the resin, squeeze flow, and resin bleed out. Figure 4 shows a highly non-linear multivariable model in which chemo-rheological resin behavior was used to determine the effective flow-compaction characteristics of the prepreg. Therefore, the proposed model was implemented in a commercially available FEA software ABAQUS/Explicit 2021 (Dassualt Systemes, Vélizy-Villacoublay, France) using custom-built subroutines. The rheological and physical state properties, such as the degree of cure, viscosity, fiber volume fraction, and fiber orientation, updated the in situ effective anisotropic viscosities of the fiber tow. The elements of the present model may be calibrated for different elastic and rheological properties for different material systems. With a large number of factors involved in the model, future algorithms, such as smoothed-FEA and machine learning, can be used to predict the deformation response of composites during manufacturing [30,31]. However, the present methodology is a more accessible approach for the process modeling of composites [32,33].

### 3.1. Chemo-Rheological Model of Thermosetting Prepreg

The degree of cure and viscosity of resin govern the squeeze flow behavior of the prepreg during the compaction process [34]. No significant exothermic reaction was observed due to the small thickness [16]. The chemo-rheological model of resin provided the degree of cure, *α*, and the resin viscosity, *μ*. The modified autocatalytic resin kinetics model were used in the present work [35]:(1)$$\frac{d\alpha }{dT}=\frac{Ae^{-\frac{∆E}{RT}}\alpha^{m_{\alpha }}{\left(1-\alpha \right)}^{n_{\alpha }}}{1+e^{C_{\alpha }\left\{\alpha -\left(\alpha_{c0}+\alpha_{CT}T\right)\right\}}}$$
where $∆E=66,500\frac{J}{mol\cdot K}$ is the activation energy, $A=153,000\frac{1}{s}$ is a pre-exponential cure rate coefficient, and *R* is the gas constant. $m_{\alpha }=0.813$ is the first exponential coefficient, $n_{\alpha }=2.74$ is a second exponential coefficient, $C_{\alpha }=4.31$ is a diffusion constant, $\alpha_{C0}=-1.684$ is the critical degree of cure at *T* = 0 K, and finally, $\alpha_{CT}=0.005473$ is a constant that governs the increase in the critical resin degree of cure with temperature, where α is the degree of cure and T is the imposed temperature. The initial degree of cure of the composite samples was measured using digital scanning calorimetry (DSC), which determined it to be 0.145.

As the resin cures, the viscosity increases exponentially. The function for the viscosity uses the degree of cure and temperature as shown in Equation (2):(2)$$\mu =\mu_{\infty }e^{k_{i}\alpha }e^{\frac{U}{RT}}$$
where $\mu_{\infty }=83\times 10^{-8}\;MPa\cdot s$ is a viscosity constant, U=49,100 kJ is the activation energy of the viscosity, and $k_{1}=25.49\frac{kJ}{mol}\;and\;k_{2}=-1.32\frac{kJ}{mol}$ are temperature-independent constants determined via the experimental data [36]. The applied temperature, the evolution of the degree of cure, and the viscosity is shown in Figure 5. The degree of cure was seen only reaching a 17% cure at the end of the hold stage, which means the resin remains in the viscous state during the experiment [37,38]. Upon gelation, the viscosity increases exponentially and the material develops solid viscoelastic properties. Therefore, the flow-compaction analysis was restricted to the first stage of the cure cycle, since the large non-linear deformation and resin flow will develop prior to gelation. Also, the chemical and thermally induced shrinkages that develop prior to gelation will be relaxed due to the liquid resin state [39,40].

### 3.2. Constitutive Squeeze Flow Model during Prepreg Fiber Tow Compaction

When heated above the instantaneous glass transition temperature, the resin becomes a viscoelsatic fluid, while rigid continuous fibers result in anisotropic viscous behavior of the uncured prepreg. Specifically, the inextensible fibers lead to the axial viscosity being six orders of magnitude higher than transverse and shear viscosity [29]. Therefore, the constitutive model simulates the unidirectional material with transversely isotropic properties. The flow-compaction model incorporated visco-hyperelastic behavior to capture the contribution of the liquid resin and included an elastic part and a viscous part [41]. The elastic part was calculated from a hyperelastic potential energy equation, ψ, using the five invariants of the Cauchy–Green deformation tensor, *C*.
(3)$$\hat{\sigma }={\hat{\sigma }}^{e}+{\hat{\sigma }}^{v}=2\left(\frac{\partial \psi^{e}}{\partial \hat{C}}+\frac{\partial \psi^{v}}{\partial \dot{\hat{C}}}\right)=2\sum_{\alpha =1}^{5}(\frac{\partial \psi^{e}}{\partial I_{\alpha }}\frac{\partial I_{\alpha }}{\partial \hat{C}})+{\hat{\sigma }}^{v}=\frac{2}{J}\left(\hat{F}\cdot \frac{\partial \psi^{e}}{\partial \hat{C}}\cdot {\hat{F}}^{T}\right)+{\hat{\sigma }}^{v}$$

Through a push-forward operation, the stress becomes a function of the deformation gradient, $\hat{F}$, and its Jacobian, *J*. The fiber direction was solved using the linear-elastic behavior from the fiber instead of the homogenized composite since it is governed by inextensible fibers. The constitutive equation for the stress is given by Equation (4) [42].
(4)$$\sigma_{11}^{e}=2\mu_{f}\varepsilon_{11}+\lambda_{f}tr(\varepsilon )$$
where $\varepsilon_{11}$ is the strain in the fiber direction and $tr\left(\varepsilon \right)$ is the trace of the strain. The constants μ and λ are related to the fiber material properties as
(5)$$\mu_{f}=\frac{E_{f}}{2\left(1+\nu_{f}\right)}$$
(6)$$\lambda_{f}=\frac{\mu_{f}\nu_{f}}{\left(1-2\nu_{f}\right)}$$
where $E_{f}$ = 168 GPa and $\nu_{f}=0.027$ are the modulus of elasticity and Poisson’s ratio of the IM7 fiber, respectively.

For the transverse and shear properties, a transversely isotropic neo-Hookean model was used to calculate the stress of the current configuration. The strain energy potential used for the 2–3 plane for the stress is shown in Equation (7).
(7)$$\psi^{e}=\frac{1}{2}\mu \left(I_{1}-3\right)-\mu \ln\left(J\right)+\frac{1}{2}\lambda {\left(J-1\right)}^{2}+\left[\beta +\zeta \ln\left(J\right)+\kappa \left(I_{4}-1\right)\right]\left(I_{4}-1\right)-\frac{1}{2}\beta (I_{5}-1)$$

$I_{1}$–$I_{5}$ are the invariants of the Cauchy–Green strain tensor. The material constants, λ, μ, β, ζ, and κ, are material constants derived from the material constants of the homogenized composites using Equations (8)–(11) [43]. The model uses the invariants of the Cauchy–Green strain tensor.
(8)$$\lambda =\frac{E_{22}\left(\nu +n\nu^{2}\right)}{m(1+\nu )}$$
(9)$$\mu =\frac{E_{22}}{2\left(1+\nu \right)}$$
(10)$$\beta =\mu -G_{23}$$
(11)$$\zeta =\frac{E_{22}\nu \left(1-n\right)}{4m(1+\nu )}$$
(12)$$\kappa =\frac{E_{11}\left(1-\nu \right)}{8m}-\frac{\lambda +2\mu }{8}+\frac{\beta }{2}-\zeta$$
(13)$$m=1-\nu -2n\nu^{2}$$
(14)$$n=\frac{E_{11}}{E_{22}}$$
where $E_{11}$ = 25 MPa and $E_{22}$ = 1 Pa are the Young’s modulus in the fiber and transverse to the fiber directions of the homogenized uncured composite to ensure the correct bending stiffness and the stability of the FEA, respectively. ν= 0.01, and $G_{23}$ = 1 MPa are the Poisson’s ratio and the shear modulus of the homogenized composite, respectively [41]. The Cauchy stress tensor can then be derived by differentiating the strain energy potential and pushing forward with a double multiplication of the deformation gradient provided by Abaqus [43]. Since the flow and the hyperelasticity constitutive equations are dependent on the fiber direction, the fiber orientation is updated every step. A model assumed that the fiber orientation follows an affine motion based on the deformation gradient characterized by the following equation [44]:(15)$$\vec{P}=\frac{\hat{F}\cdot \vec{p_{0}}}{\left\|\hat{F}\cdot \vec{p_{0}}\right\|}$$
where $\vec{p}$ is the fiber direction vector and $\vec{p_{0}}$ is the initial fiber direction vector. The fiber direction is used to inform the anisotropic component of the hyperelastic model, where it becomes the variable for the function of stress based on the strain energy potential [24].

The microscopic fibers suspended in resin show time-dependent behavior based on the strain rate effects and resin viscous behavior. Research from Kelly [28] used the multiplicative decomposition of apparent viscosity to describe the squeeze flow. To quantify the homogenized prepreg tape response, the proposed approach for the viscous term uses multiplicative components that consist of strain- and strain rate-dependent terms [28,29,41]. The strain dependent terms take into account the ply geometry and the strain rate described as the resin behavior [41]. The viscous stress term followed the behavior as shown in Equation (16).
(16)$${\sigma^{v}}_{ij}={\eta_{rate}}_{ij}\cdot {\eta_{ply}}_{ij}\cdot {\eta_{micro}}_{ij}\cdot {\hat{\dot{\varepsilon }}}_{ij}$$
where ${\hat{\eta }}_{rate}$ is a strain rate-dependent viscosity tensor component and $\hat{\dot{\varepsilon }}$ is the strain rate tensor. ${\hat{\eta }}_{ply}$ is the term responsible for ply geometry and ${\hat{\eta }}_{micro}$ is the phenomenological strain-dependent term derived from micromechanical considerations at the micro level.

During the curing, the fiber–resin system is assumed to act as a power-law viscous fluid. Previous work described the anisotropic viscosity of oriented discontinuous fibers suspended in Newtonian fluid [29]. The same principles may be applied to continuous fiber with a finite length. Therefore, a tensorial viscosity approach was used to model the ${\hat{\eta }}_{rate}$ term where the resin matrix follows a power-law behavior. Since the composite tow is transversely isotropic, the in-plane directions (transverse to the fiber) were assumed to have the same properties, while the fiber direction is assumed to be inextensible.

In Nixon’s work [23], the strain rate term used phenomenological parameters that were determined experimentally. The proposed rate term for the current study uses effective properties of the composite derived from the anisotropic viscosity as a function of the strain rate, fiber volume fraction, and a power-law exponent using the following tensor components [29].
(17)$${\eta_{rate}}_{11}=\eta_{resin}2^{-mf}{\left[\frac{\sqrt{\bar{f}}}{1-\sqrt{\bar{f}}}\right]}^{m}{\left(\frac{L}{D}\right)}^{m+1}{{\dot{\epsilon }}_{11}}^{m-1}$$
(18)$${\eta_{rate}}_{22}={\eta_{rate}}_{33}=\eta_{resin}{2^{m+1}\left[1-\sqrt{\bar{f}}\right]}^{-m}{{\dot{\epsilon }}_{22}}^{m-1}$$
(19)$${\eta_{rate}}_{23}=\eta_{resin}{\left[1-\sqrt{\bar{f}}\right]}^{-m}{{\dot{\gamma }}_{23}}^{m-1}$$
(20)$${\eta_{rate}}_{12}={\eta_{rate}}_{13}={\eta_{resin}2}^{-m}{\left[\frac{1-\sqrt{\bar{f}}}{1-\sqrt{\bar{f}}}\right]}^{m}{{\dot{\gamma }}_{12}}^{m-1}$$
where $\bar{f}$ is the fiber volume fraction, *L* is the fiber length, *D* is the fiber diameter, and *m* = 0.5 is the power-law exponent.

The ply geometry term is derived from an analytical solution by Kelly [28] and describes the geometry and surface conditions of the ply. A frictionless boundary condition was assumed for the solution since the scope of the curing and compaction has the resin as a fluid. A solution to the work, rearranged by Belnoue et al. [41], was used in this study, as shown in the equations below.
(21)$$\eta_{ply_{11}}={\hat{\eta }}_{ply_{12}}={\hat{\eta }}_{ply_{13}}={\hat{\eta }}_{ply_{23}}=1$$
(22)$${\eta_{ply}}_{22}=2{\left(\frac{w_{0}}{h_{0}}\right)}^{2}e^{-4\varepsilon_{22}}$$
(23)$${\eta_{ply}}_{33}=2{\left(\frac{w_{0}}{h_{0}}\right)}^{2}e^{-4\varepsilon_{33}}$$
where *w*_0_ is the initial width of the ply, and *h*_0_, is the initial height of the ply.

The micro term depends on the phenomenological parameters that govern the flow regimes between an incompressible solid to a compressible solid due to resin bleed out in the fiber direction as a result of fiber locking during compaction [45].
(24)$$\eta_{micro_{11}}=\eta_{micro_{12}}=\eta_{micro_{13}}=\eta_{micro_{23}}=1$$
(25)$$\eta_{micro_{22}}=2\sqrt{\chi_{l}}e^{\varepsilon_{22}}k\left({\left(\frac{k}{\sqrt{\chi_{f}}e^{\varepsilon_{22}}-k}\right)}^{2}+3\right)$$
(26)$$\eta_{micro_{33}}=2\sqrt{\chi_{l}}e^{\varepsilon_{33}}k\left({\left(\frac{k}{\sqrt{\chi_{f}}e^{\varepsilon_{22}}-k}\right)}^{2}+3\right)$$
where *k* is a stepwise function of the temperature, $\chi_{l}=0.63$ is the aspect ratio during locking, and $\chi_{f}=0.785$ is the final aspect ratio based on the maximum packing fraction of the composite [46]. The packing in this case is interpreted as the fiber volume fraction.

### 3.3. Finite Element Modeling of Compaction in Unidirectional Tows and Fiber Tow Gap Samples

The proposed flow-compaction constitutive behavior was validated using the experimental results from unidirectional fiber tows with varying widths and thicknesses (Figure 6a). The same geometry was modeled and simulated using FEA with the proposed visco-hyperelastic behavior (Figure 6b). There were 60 to 200 C3D8T elements depending on the width of the composite, with one element through the thickness for all configurations. The model was constrained in the *z*-direction to simulate the rigid tool used in the autoclave curing process. The strain was measured using the average deformation of the nodes on the top of the ply.

The tow gap configuration was used for understanding the effect of resin bleed out on the cured morphology around the gap region and used the same stacking sequence and geometry as the experimentally manufactured composite (Figure 7). To incorporate the initial sinking of the top 0° ply created via a manually applied roller during the layup, the initial slope was incorporated by connecting the bottom and top 0° plies at the middle of the gap. Symmetrical boundary conditions were used on the *x*-*z* plane of the 0° plies towards the gap and one side towards of the *y*-*z* plane to reduce processing time. To simulate the tool, the bottom of the laminate model was constrained in the *z*-direction. The model consisted of 2088 C3D8RT elements to capture the non-linear deformation. Frictionless contact between the layers of 0° and 90° plies was used. Each ply had a nominal 0.14 mm in thickness with five elements through the thickness. The models used a density for the homogenized composite of 1590 kg/m^3^ with a mass scaling of 10 to reduce the amount of time to complete the FEA simulations. No significant rise in kinetic energy was found with the addition of the mass scaling. The material properties for both the IM7 carbon fibers and the 8552 resin were derived from the literature and are utilized in the subroutines [19,35,36]. Geometric non-linearity was activated for both models to accommodate the large deformations during the compaction.

## 4. Results

### 4.1. Experimental Quantification of Resin Bleed Out in Unidirectional Prepreg Tows

Six configurations with six samples each of the unidirectional composites were compacted and cured in the first 90 min of the cure cycle (Figure 5) A tow aspect ratio in unidirectional tow was introduced based on the number of tows in the width and through the thickness:(27)$$\frac{N_{w}}{N_{T}}=\xi$$
where *N_w_* is the number of plies in the tow width (transverse of the fiber direction), *N_T_* is the number of tows through the thickness of the composite, and ξ is the geometric aspect ratio. For example, a sample with a thickness of four tows and a width of one tow had ξ = 0.25. The experimental samples and corresponding simulations used the proposed fiber tow aspect notation.

Resin bleeds out of the porous fiber bed when heat and pressure are applied [47]. This resin bleed out is a complex process which depends on resin viscosity, local fiber volume fraction, applied autoclave pressure, and pressure distribution throughout the laminate. In the present study, a phenomenological approach was proposed which considered the amount of resin bleed out as a function of time for the given cure cycle. The experimental results of unidirectional tow compaction demonstrated that resin bleed out develops at the initial stages of the manufacturing process where the flow was considered separately in the longitudinal and transverse to the fiber directions. The amount of resin that came out in the respective directions were cut from the composite and weighed. The mass loss in each direction was calculated as a ratio of the measured resin mass.

The average bleed-out mass per unit of surface area, based on the initial dimensions, was measured. While the mass loss for the transverse resin weighs more, when the corresponding area is considered, the resin flowing longitudinally along the fiber direction was responsible for a larger fraction of mass loss per surface area. Longitudinal bleed out occurred through the surface as normal to the fibers, while transverse flow occurred on the side of the prepreg tape—no transverse flow occurred due the tool surface on the bottom and the release film on top. Greater resin loss in the longitudinal direction corresponds to the higher permeability in the fiber direction compared to permeability in the transverse direction [48]. The single tow configuration (ξ = 1.0) shows that the longitudinal direction bleeds out more resin mass per unit area, with a coefficient ${{\bar{\rho }}_{P}}^{\left(L\right)}$ = 0.400 mg/mm^2^, compared to the transverse direction, with a mass loss coefficient ${{\bar{\rho }}_{P}}^{\left(T\right)}$ = 0.0198 mg/mm^2^ at 90 min into the cure cycle. The area of resin around a single tow (Figure 2b) was measured transversely and longitudinally over time and used as a scale to estimate the mass loss over time per unit area over time (Figure 8). Functions representing the mass loss coefficient over time in longitudinal, ${{\bar{\rho }}_{P}}^{\left(L\right)}(t)$, and transverse, ${{\bar{\rho }}_{P}}^{\left(T\right)}(t)$, directions were based on the logistic curve functions in Equation (28). Note that only the transverse resin bleed out was present around the surface area of the gap (Figure 3).
(28a)$${{\bar{\rho }}_{P}}^{\left(T\right)}(t)=\frac{{A^{\left(T\right)}}_{1}}{\left(1+e^{\frac{{A^{\left(T\right)}}_{2}-t}{{A^{\left(T\right)}}_{3}}\;}\right)}$$
(28b)$${{\bar{\rho }}_{P}}^{\left(L\right)}(t)=\frac{{A^{\left(L\right)}}_{1}}{\left(1+e^{\frac{{A^{\left(L\right)}}_{2}-t}{{A^{\left(L\right)}}_{3}}\;}\right)}$$
where *A*^(*T*)^_1_ = 0.0195 mg, *A*^(*T*)^_2_ = 33.542 min, and *A*^(*T*)^_3_ = 3.10 min are transverse fitting parameters and *A*^(*L*)^_1_ = 0.0293 mg, *A*^(*L*)^_2_ = 43.881 min, and *A*^(*L*)^_3_ = 11.78 are the longitudinal parameters.

The experimental data shows that the resin begins flowing between 20 and 30 min into the cure cycle before plateauing due to decreased permeability and increased viscosity. The resin mass loss data was also used to quantify the fiber volume fraction as a function of the through-thickness strain. An average initial value of the fiber volume fraction of 0.55 was derived from thermogravimetric (TGA) analysis [49]. The initial weight of the resin was determined using the matrix volume fraction based on TGA analysis, and the known densities of the matrix and the fiber was determined via the rule of mixtures. The change in width and length and the known initial volume of a single unidirectional tow (ξ = 1.0) was used to calculate the through-thickness strain, $\varepsilon_{33}$ over time. This allowed to fit the fiber volume with respect to the compaction strain as shown in Figure 9 as an exponential plateau:(29)$$V_{f}\left(\varepsilon_{33}\right)=V_{f_{0}}+B_{1}*e^{\left(\frac{-B_{2}}{\varepsilon_{33}}\right)}\;$$
where $B_{1}=0.0673\;and\;B_{2}=0.0084\;$ are fitting parameters, $\varepsilon_{33}$ is the strain in the compaction direction, and $V_{f_{0}}$ is the initial fiber volume fraction. The fiber volume fraction rapidly increased at the early stages of compaction and plateaus as the permeability decreased and resin viscosity increased. Furthermore, the resin bleed out present at the beginning of the cure cycle influenced the squeeze flow via the evolution of the fiber volume fraction, which affects the anisotropic apparent viscosity in Equations (17)–(20). The fiber volume fraction formulation was measured for the prescribed cure cycle and material system. The same methods can be recreated with different temperature profiles and composite materials. With the resin bleed out and the development of the fiber volume fraction for the current cure cycle established, a single-phase model was made where the resin is not explicitly modeled. Modeling the flow of resin explicitly can be achieved with a higher resolution model. However, this would require computationally expensive representative volume elements to model the microscopic behavior. The present analytical equation allows for an adequate estimate of the evolution of the fiber volume fraction.

### 4.2. Modeling and Validation of Compaction Behavior in Unidirectional Tows

The compaction results of the unidirectional tows showed that the thicker samples display more tow spreading in comparison to the tows with a higher width. For a configuration of a single tow (ξ = 1), the maximum compaction strain was shown to be 0.053 at 90 min into the cure cycle. The largest compaction strain, *ε*_33_ = 0.137, was found for the four tows arranged in the thickness direction (ξ = 0.25), while the smallest strain *ε*_33_ = 0.012 was found for the four tows arranged in the width direction (ξ = 4.0). The range in the observed compaction strain shows the importance of the tow geometry, and more specifically, the thickness of the ply, on the squeeze flow behavior.

The undeformed and deformed configuration of the unidirectional models are shown in Figure 10. The baseline configuration for the unidirectional models with one ply (*ξ* = 1.0) shows the thickness decreased and the tow spread at the Y-direction (transverse to the fibers). The modeling results in terms of the through-thickness compaction strains were compared with the experimental measurements to validate the proposed flow-compaction model. The same trend between the experimental and numerical results demonstrated that the transverse spreading increased, with more plies through the thickness, and decreased with more tows across the width.

The through-thickness strains, *ε_33_*, in the FEA models of unidirectional tows were quantified using the average displacement on the top *y*-*x* surface and the known initial thicknesses. Figure 11a shows the rapid decay of the through-thickness strain, *ε*_33_, at 30 min into the hold stage with respect to the aspect ratio. The tow thickness during the compaction simulations and the experimental measurements followed a non-linear visco-hyperelastic curve and decreased with the aspect ratio (Figure 11b). Both the experimental and modeling prediction of compaction agreed closely that up to 90 min of the hold stage was observed, and deviation was noticed at the end of the hold stage. The inclusion of updated fiber volume fraction as a function of the compaction strain (Equation (29)), due to resin bleed out, resulted in improved accuracy of the tow compaction strain (Figure 11a).

### 4.3. Squeeze Flow Simulation of Layup with Embedded Tow Gap

To further validate the proposed compaction constitutive model, the region around the fiber tow gap was analyzed. The initial undeformed configuration of the fiber tow gap model is shown in Figure 12a, which demonstrates the layup compaction during the simulation. The initial FEA simulation did not consider coupling between the squeeze flow and resin bleed out, which is discussed in the following section. The top ply sank into the fiber tow gap, creating a curvilinear triangular void near the edge of the 90° ply. Early in the cure cycle (before 13 min), the gap closed rapidly until the autoclave pressure reached the maximum 0.541 MPa (80 psi). After the initial gap closure, the gap volume decreased slowly until the end of the isothermal hold stage. Tow spreading was observed at the edge of the 90° ply.

The simulation results with only squeeze flow were compared to the experimentally manufactured sample with the fiber tow gap. The micrograph of the area of interest showed the morphology due to the fiber tow gap in Figure 12b, in which the top 0° ply sank into the gap. The resin-rich region was formed because of resin bleed out from the adjacent plies. The experimental size of the resin-rich region was significantly larger (0.811 mm in length) compared to the one predicted from squeeze flow analysis (0.227 mm in the closed gap), as shown in Figure 12. A much steeper slope in the top 0° ply was more pronounced in the simulation than in the physical piece. The explanation for this discrepancy in the predicted and observed tow morphology is due to resin bleed out, which provides the support against the sinking to a 0° ply. This result illustrates that resin bleed out must be considered to capture the cured defect morphology.

### 4.4. Effect of Resin Bleed Out during Compaction of Layup with Embedded Tow Gap

The squeeze flow simulation of the fiber tow gap was used to track the changes in the gap volume. The size of the void in squeeze flow analysis was shown to rapidly decrease because of top ply sinking, where the void volume quickly dropped from 0 to 13 min and then plateaued. However, it is evident that resin bleed out occurs simultaneously with the squeeze flow in the early stages of the cure cycle (see Figure 8), which slows down the volume of the gap from decreasing. In order to capture the coupling between the two phenomena, the squeeze flow, and the resin bleed out, the following procedure was used. The volume of the bled-out resin was estimated using the following equation:(30)$$V_{Res}\left(t\right)=\frac{{{\bar{\rho }}_{P}}^{\left(T\right)}\left(t\right)\cdot \;SA_{Gap}}{\rho_{r}}\;$$
where the density of resin, $\rho_{r}$*,* was 1100 g/cm^3^; $SA_{Gap}$ is the initial surface of the fiber tow gap volume *V_Gap_*(*t*), which changes during the compaction; and ${{\bar{\rho }}_{P}}^{\left(T\right)}$ is the experimentally determined resin bleed-out areal density for the prescribed cure cycle, as given by Equation (28a).

Over time, the volume of the gap region, *V_Gap_*(*t*), is expected to close, as a result of ply sinking and squeeze flow, while at the same time, the volume of the bled-out resin is increasing, *V_Res_*(*t*). Once the volume of the bled-out resin reaches the volume of the tow gap void where the resin-rich region is formed. The time of gap filling with resin was predicted to occur from the squeeze flow simulation at *t_fill_ =* 26 min (Figure 13). On the other hand, the estimated experimental volume of the resin pocket in the micrograph is shown as a horizontal line in Figure 13. The volume of the resin-rich pocket was found from a micrograph in Figure 12b and corresponded to the volume of resin in the gap at a slightly later time of 29 min, which indicates the progressive filling and coupling between the squeeze flow and resin bleed out.

The following method was used to describe this flow coupling effect. A pressure, matching the autoclave pressure, was applied on the internal surfaces of the closing gap to capture the presence of bled-out resin as it fills the void. Furthermore, to simulate the progressive filling of the gap, the internal void pressure was activated gradually in partitioned regions 1–4, as shown in Figure 12a, until the predicted moment in time, *t_fill_*, when the gap void is supported by the resin. The internal pressure was enabled, starting from the symmetrical end of the model, which was selected based on the proximity of the void surfaces in that region. The pressure was activated at 6, 12, 18, and 24 min, respectively. To capture the stress relaxation effects in the viscoelastic liquid resin, the applied pressure followed a linear decrease over 25 min (Figure 12a). The time was selected to correspond to the overall time of filling of the tow gap region. With the addition of the pressure, the initial sinking of the fiber tow gap was slowed. With the incorporation of the coupling between squeeze flow and resin bleed out, the void filling was predicted to occur at a later time than in squeeze flow analysis (28 min compared to 26 min), which was closer to the moment corresponding to the predicted time of filling (at 29 min), based on the experimental size of the resin-rich pocket (Figure 13). In conjunction with the change in the fiber volume fraction, the resin bleed out was calibrated to the specific material and cure history. The sequential analysis and modeling method presented was calibrated for the current configuration but can be used on other defect features.

### 4.5. Evaluation of Morphology and Fiber Volume Fraction in Cured Layup with Fiber Tow Gap

With the bled-out resin providing structural support against the sinking ply above the tow gap, the final morphology of the fiber tow gap model with the internal void pressure is shown in Figure 12a. When comparing the two FEA models (squeeze flow with and without the effect of resin bleed out) at 13 min, the void pressure applied prevented the ply from sinking too rapidly. The out-of-plane waviness for the top 0° ply, during the hold stage (45 min), was more gradual and developed over a larger distance, which matched the experimental observation. The out-of-plane angle measured in the top 0° ply is shown in Figure 14, where the simulation captured the evolution of waviness along the length of the gap; however, the experimental results were not as steep as the modeling prediction. The size of the predicted resin-rich region from the coupled simulation had a length of 0.946 mm (compared to the 0.227 mm without the internal void pressure), which agreed with the experimental size of 0.811 mm. The thickness variation in the bottom 0° ply is shown in Figure 15. The measured thickness in the micrograph was found to increase from the left of the reference line towards the center of the composite. In comparison, the numerical model followed a curve which had the thickest part at the reference line and decreased in both directions. The resulting morphology can be utilized in further analysis of the composite, such as through progressive damage analysis, to compare with fabricated laminates in the future [50].

Predictions of the variable volume fraction of the 90° ply is shown in Figure 16a. A micrograph of the center 90° ply from the cured sample was discretized into rectangular sections where the fiber volume fraction, *V_f_*, was quantified. The micrograph showed a *V_f_* range of 0.40–0.57 with a resin-rich region in the top corner that had a *V_f_* of about 0.10. FEA simulations predicted *V_f_* to a range from 0.60 to 0.70. This study did not consider the initial distribution of the local fiber volume fraction that can account for some of this variation.

## 5. Conclusions

The interconnected multiphysics framework, which incorporated chemo-rheological and coupled flow-compaction analyses, was developed to simulate the compaction of AFP prepreg preform in order to predict complex morphology around the region of deposited fiber tow gap. Anisotropic tensorial prepreg viscosity was used to capture tow spreading during the cure cycle. The deformation in unidirectional and fiber tow gap samples was observed to be a result of the coupling between the squeeze flow and resin bleed out. Resin bleed out was quantified by measuring the volumetric flux of the resin from unidirectional prepreg tape. The bled-out resin was found to fill the void in the samples with a single fiber tow gap. To capture this phenomenon in simulation, sequential compaction-flow analysis was proposed and validated. The multi-physics modeling approach connects the squeeze flow simulation with the effect of resin bleed out through the evolution of void closure in the simulation. As the first step, squeeze flow simulation is used to predict the evolution of the gap and estimate the time at which the void is filled with resin. At the second step, the calculated time of void filling was used to activate the internal pressure inside of the void. Instead of modeling the resin explicitly, a one-phase model was proposed, which considers the effect of resin bleed out via the activated internal void pressure. The void pressure is activated progressively to simulate the void filling with resin based on measured resin bleed-out behavior in unidirectional tows. By restricting early sinking of the top ply into the gap, the proposed modeling approach resulted in the cured morphology that captured the non-uniform morphology more accurately around the initial tow gap region.

The modeling framework demonstrated that morphological features develop due to the coupled manufacturing phenomena and cannot be considered in isolation from each other. The phenomena include the chemo-rheology of the resin and visco-hyperelasticity of the laminate in the early stages of curing. The compaction strain was connected to the local fiber volume fraction via tensorial viscosity and it allowed us to consider the interaction between resin bleed out and squeeze flow. The initial deposition gap regions develop into the non-uniform morphological features in the composite, which includes the non-uniform ply thicknesses, fiber waviness, resin-rich regions, and the fiber volume fraction variation. Moreover, the proposed multi-physics modeling framework can be adopted to accommodate various material systems with different chemo-rheological properties. The proposed sequential modeling approach has been used to predict cured morphology in more complex layups with spatially distributed, staggered fiber tow gaps [51]. The proposed modeling approach can be used to decouple the effect of various effects on the cure morphology in AFP composites, including the material uncertainty, such as initial fiber volume fraction and fiber orientation distributions.

## Figures and Tables

**Figure 1 polymers-16-00031-f001:**
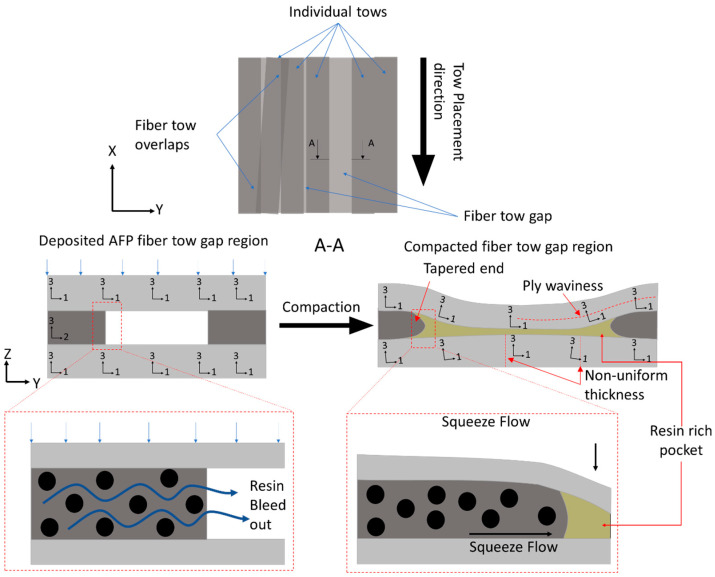
Schematics of gaps within a laminate and the effects of resin bleed out on the non-linear morphology.

**Figure 2 polymers-16-00031-f002:**
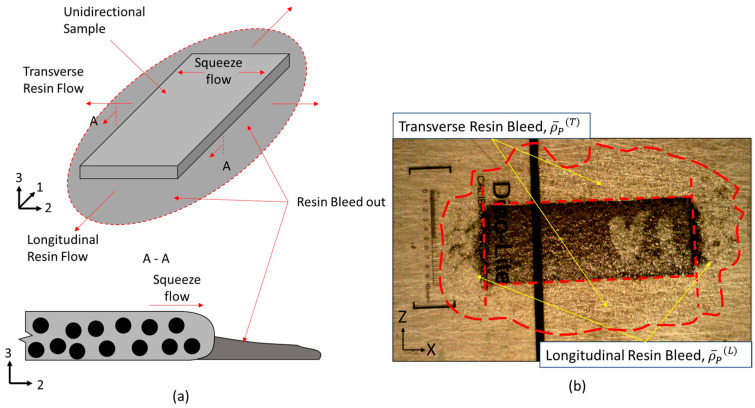
(**a**) Schematics of the bleed out in a unidirectional composite sample. (**b**) A unidirectional sample which shows a dashed red line dividing the transverse resin bleed, longitudinal resin bleed and the area of the unidirectional tow.

**Figure 3 polymers-16-00031-f003:**
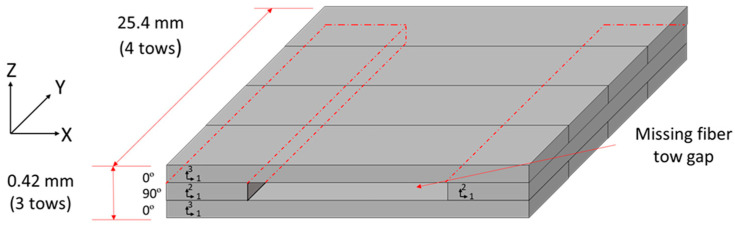
Schematics of the manufactured fiber tow gap dimensions and stacking sequence. The dashed line indicates the 90º ply between the 0º plies in the laminate.

**Figure 4 polymers-16-00031-f004:**
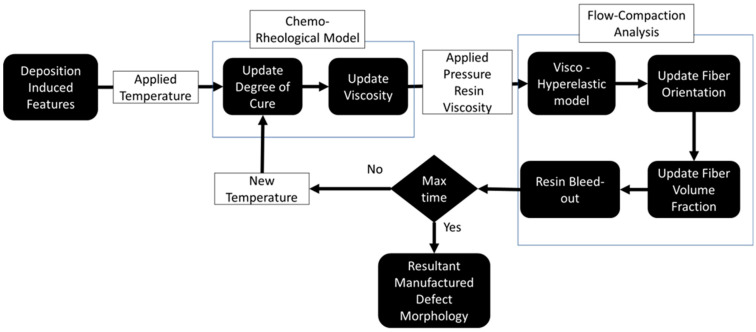
Coupled chemo-rheological and compaction model for predicting defect morphology in AFP-manufactured composites.

**Figure 5 polymers-16-00031-f005:**
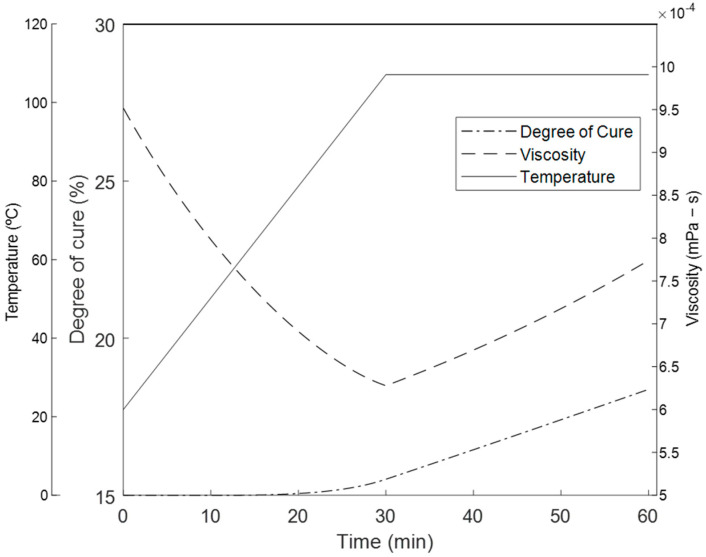
The applied temperature and the evolution of the degree of cure and viscosity using the thermo-rheological model.

**Figure 6 polymers-16-00031-f006:**
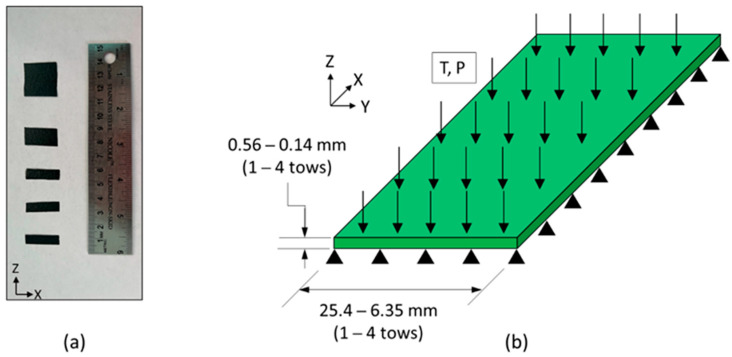
(**a**) The fully cured samples of the unidirectional experiments and the aspect ratio in order from left to right and labeled with the aspect ratio (ξ): one ply thick (ξ = 1.0), two ply thick (ξ = 0.5), four ply thick (ξ = 0.25), two ply wide (ξ = 2.0), and four ply thick (ξ = 4.0) (**b**) Schematics of the unidirectional FEA models, where it was varied by the same aspect ratios. (**b**) Schematics of the unidirectional experiments, where the width and thicknesses were varied.

**Figure 7 polymers-16-00031-f007:**
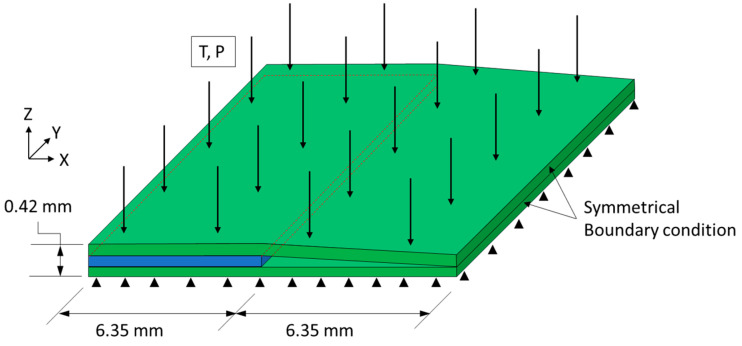
The schematics of the fiber tow gap model with boundary conditions, applied pressure The green plies indicate 0° and the blue ply indicates 90°. Internal resin pressure is applied to the gap in the middle of the 0° plies.

**Figure 8 polymers-16-00031-f008:**
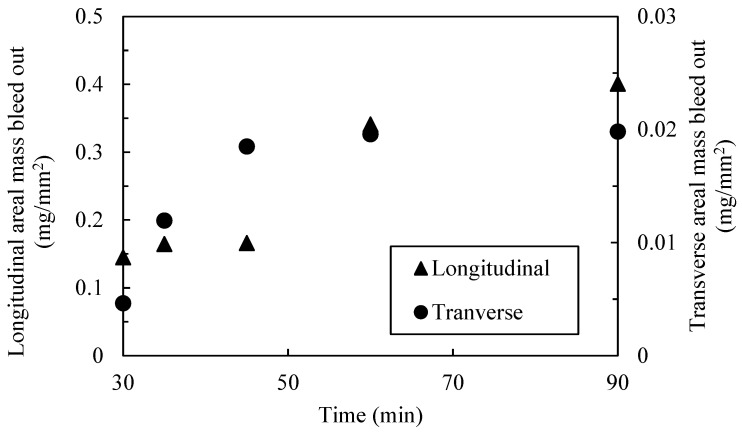
An estimated mass loss fraction over time of the fiber tow gap using experimental data from unidirectional experiments.

**Figure 9 polymers-16-00031-f009:**
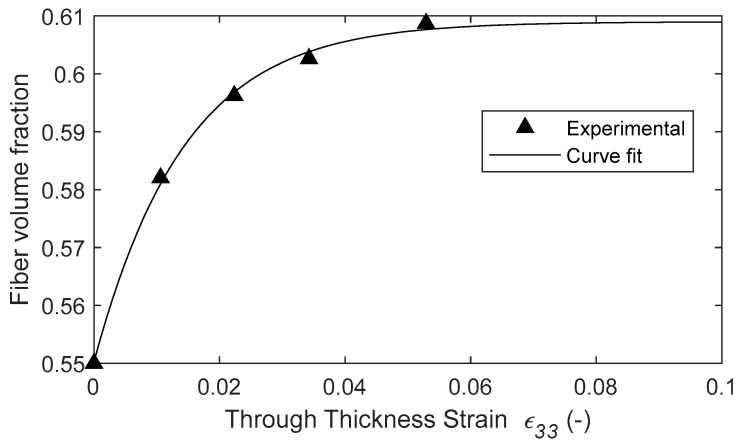
The fiber volume fraction plotted with respect to the strain.

**Figure 10 polymers-16-00031-f010:**
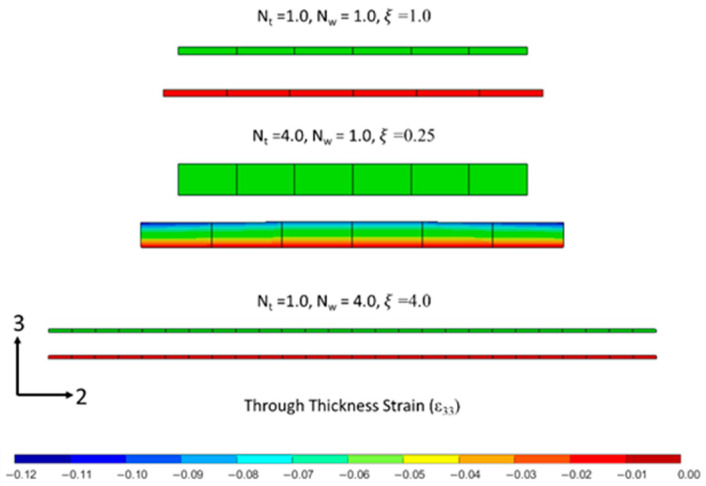
Undeformed and deformed configuration of tows from compaction simulation: a one-tow wide and thick composite (ξ = 1.0), a four-tow thick composite (ξ = 0.25), and a four-tow wide composite (ξ = 4.0).

**Figure 11 polymers-16-00031-f011:**
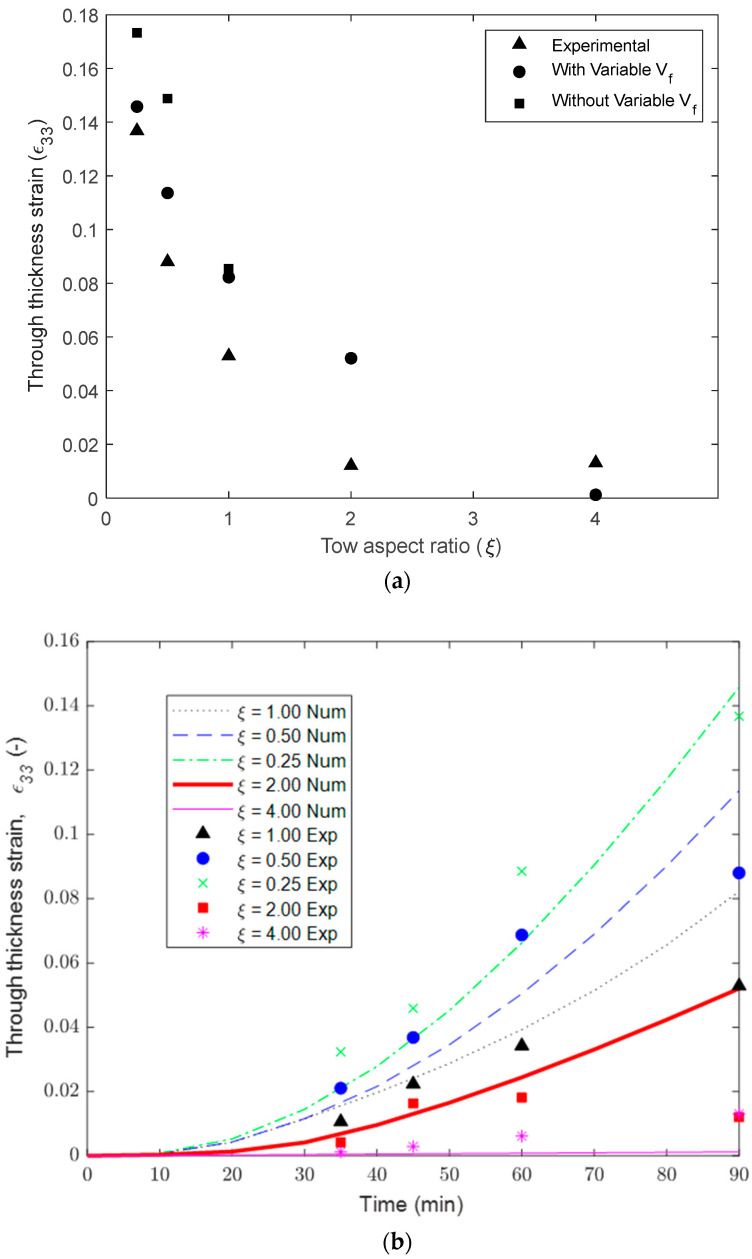
(**a**) Comparison of compaction strain in tows with different aspect ratios. (**b**) Compaction strain evolution during cure time in tows with different aspect ratios.

**Figure 12 polymers-16-00031-f012:**
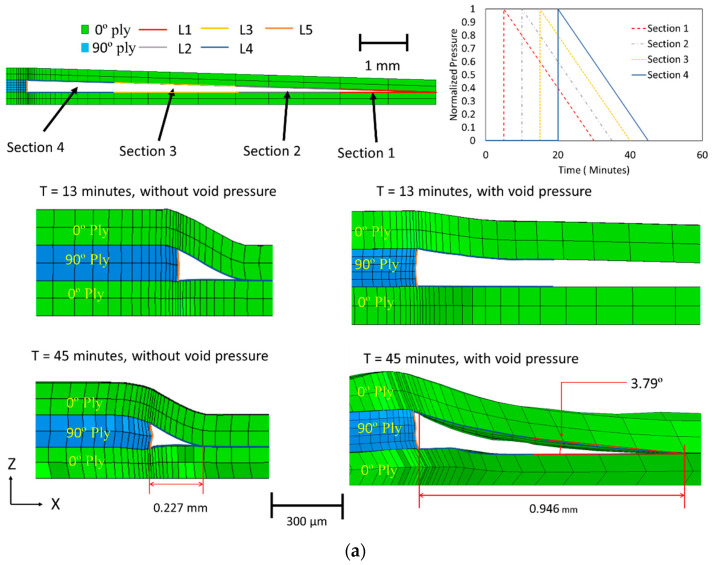
(**a**) The effect of including internal pressure on the size of the void region. (**b**) The geometry of the resin rich region around the initial tow gap in cured composite.

**Figure 13 polymers-16-00031-f013:**
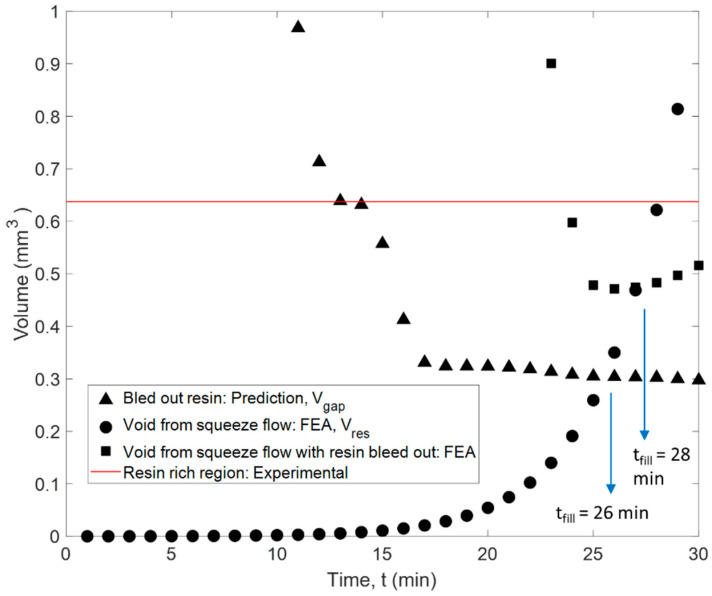
The measured volumes of the model, a theoretical mass flow volume, and the experimental gap volume measured from the micrograph.

**Figure 14 polymers-16-00031-f014:**
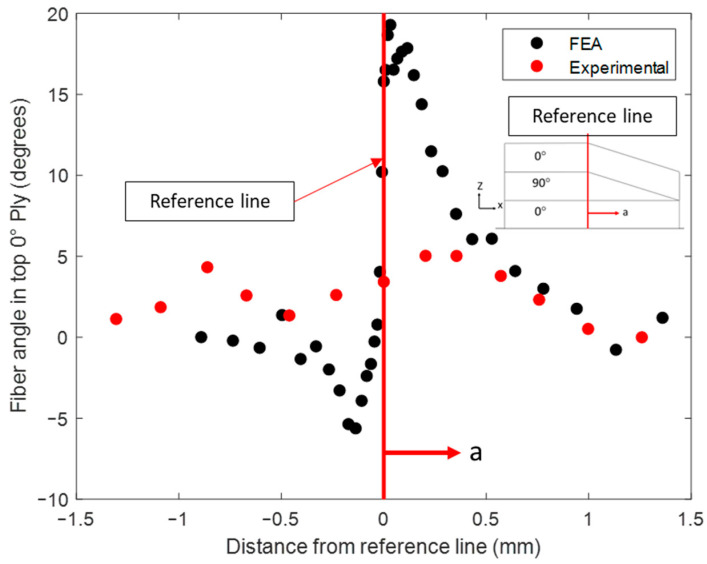
The out-of-plane fiber waviness in the top 0° ply.

**Figure 15 polymers-16-00031-f015:**
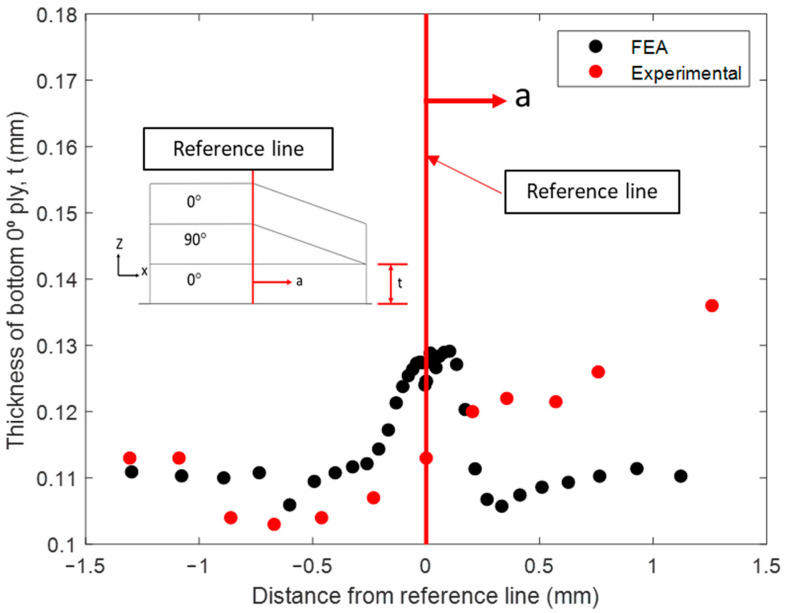
The bottom 0° ply thickness variation.

**Figure 16 polymers-16-00031-f016:**
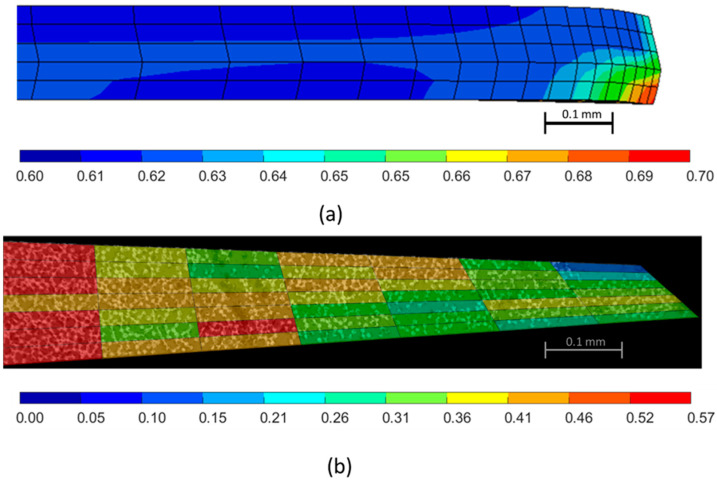
(**a**) The distribution of the fiber volume fraction in the fiber tow gap simulation of the 90° ply. (**b**) The micrograph based measurement of the fiber volume fraction of the 90° ply.

## Data Availability

The data presented in this study are available on request from the corresponding author. The data are not publicly available due to the proprietary model.

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
