# Peer review of "Effect of Resin Bleed Out on Compaction Behavior of the Fiber Tow Gap Region during Automated Fiber Placement Manufacturing"

_polymers, 2023, doi:10.3390/polym16010031_

Round 1

Reviewer 1 Report

Comments and Suggestions for Authors

An interconnected multiphysics framework was developed to simulate the compaction of thermosetting prepreg in order to predict complex morphology. Anisotropic tensorial prepreg viscosity was used to capture tow spreading during the cure cycle. The deformation in unidirectional and fiber tow gap samples was observed to be a result of the coupling between the squeeze flow and resin bleed-out. The research of the paper has a certain guiding role for practical engineering problems. 

Abbreviations are not recommended for the title.

Author Response

Reviewer Response: An interconnected multiphysics framework was developed to simulate the compaction of thermosetting prepreg in order to predict complex morphology. Anisotropic tensorial prepreg viscosity was used to capture tow spreading during the cure cycle. The deformation in unidirectional and fiber tow gap samples was observed to be a result of the coupling between the squeeze flow and resin bleed-out. The research of the paper has a certain guiding role for practical engineering problems.

Response: Thank you for reviewing my work on the process modelling of composite. and I am glad you found the paper informative. The title has been changed to avoid abbreviations.

Reviewer 2 Report

Comments and Suggestions for Authors

Review Report

Manuscript title

Effect of Resin Bleed-out on Compaction Behavior of Fiber 2 Tow Gap Region during AFP Manufacturing

Authors

Von Clyde Jamora, Virginia Rauch , Sergii G. Kravchenko and Oleksandr G. Kravchenko

Fiber tows are fundamental to the performance and versatility of polymer composites. The orientation of fiber tows introduces anisotropy to the composite material, meaning that its properties vary depending on the direction of measurement. This property is valuable in engineering applications where various parts of a structure or component experience varying loads and stresses.

Fiber tows allow for control over the orientation of the reinforcing fibers within the composite. By strategically placing and aligning the tows, engineers can tailor the material's properties to be stronger in specific directions. This directional control is crucial for designing materials that can withstand specific types of loads or stresses.

The manuscript describes  a multi-physics modeling approach based on sequential simulation of the flow-compaction, and the squeeze flow simulation used to capture the evolution of void filling.

Squeeze flow simulation allowed to estimate the time of filling of the void with resin. Activation of the internal pressure inside of the void was through the time of void filling. Thus, the cured morphology captured the non-uniform morphology more accurately around the initial tow gap region.

The proposed methodology tries to couple the phenomena of squeeze flow and the resin bleed out, which determine the properties of the composite material.

The study offers a qualitative analysis of the progress of the AFP manufacturing process, resulting in a series of conclusions regarding the influence of characteristic factors on the mechanical parameters of the composite material.

Although the study is of interest and contains an original approach, two aspects should clarify in addition, as follows.

1.       It is not clear how “automated fiber placement manufacturing method” allows for “precise control over the complex layup design,” and how the present study could enhance the process.

2.       Furthermore, the study does not show how the results obtained in this study “can be used to predict cured morphology in more complex layups with spatially distributed fiber tow gaps.”

Collateral remarks refer to the title of paragraph 4.5, where it is not about “microstructure,” and to the phrase contained in lines 496-498, which, being confusing, should be reformulated.

Author Response

  1. It is not clear how “automated fiber placement manufacturing method” allows for “precise control over the complex layup design,” and how the present study could enhance the process.

Response: Thank you for the response to the paper, and to acknowledge the sentence, automated Fiber Placement allows for tailored variable stiffness structures as outlined in the reference. Examples of how this process can be used for complex layup design has been added to the introduction to support the statement. Since gaps and overlaps are inevitable to a certain degree, the effects of the gaps and overlaps can be hard to predict without physical testing of the part. The model aims to predict the morphology of the composite, where the resultant geometry can then be used to predict the strength knockdown of the defect type. This can be utilized in the manufacturing process to predict the effects of unavoidable gaps and overlaps in complex composite layups. The sentence has been changed as a general introduction to AFP to avoid confusion. To clarify the statement, the sentence has been edited and an example of the capabilities of AFP was added:

“Automated fiber placement is a state-of-the-art manufacturing method which allows for precise control over layup design.”

“For example, Wu at al. created a variable stiffness structure with a cut-and-restart functionality of an AFP machine, to manufacture a prototype of an airplane fuselage [3].”

  1. Furthermore, the study does not show how the results obtained in this study “can be used to predict cured morphology in more complex layups with spatially distributed fiber tow gaps.”

Collateral remarks refer to the title of paragraph 4.5, where it is not about “microstructure,” and to the phrase contained in lines 496-498, which, being confusing, should be reformulated.

Response: The section title has been changed to a be more specific contained in the paragraph. Assuming the reviewer meant lines 396-398, the line has been clarified. The sentence summarizes how the interconnected modeling works and its connection to the morphology of the laminate due to the deposition features. A more thorough explanation of what the interconnected physics and features resulting form the manufacturing has been outlined in the conclusion:

“The modeling framework demonstrated that morphological features develop due to the coupled manufacturing phenomena and cannot be considered in isolation from each other. The phenomena include the chemo-rheology of the resin, and visco hyperelasticity of the laminate in the early stages of curing. The initial deposition gap regions develop into the non-uniform morphological features in the composite, which includes the non-uniform ply thickness, fiber waviness, resin rich regions, and the fiber volume fraction variation. “

Reviewer 3 Report

Comments and Suggestions for Authors

I would like to congratulate all the authors for the research work they have given to the scientific community. Kindly address my review questions, so that the manuscript can be clear for the readers

Manuscript Title - Effect of Resin Bleed-out on Compaction Behavior of Fiber Tow Gap Region during AFP Manufacturing

I am writing to express my heartfelt appreciation to all authors for your remarkable research. Your work exemplifies the diligent pursuit of scientific excellence and the quest for sustainable solutions in a specified area. For the betterment of the manuscript, Kindly address the following review questions, So that it will reach the maximum for the scientific community.

Questions

1.     Given the complexity of large nonlinear deformation in automated fiber placement (AFP), what novel computational techniques or algorithms beyond finite element methods could be explored to address the computational challenges associated with simulating and predicting such deformations accurately?

2.     Considering the interconnected nature of morphological features like non-uniform ply thickness, fiber waviness, and variable fiber volume fraction, how can the model be extended to dynamically simulate the evolution of these features over multiple manufacturing cycles or in scenarios with rapidly changing manufacturing conditions?

3.      Morphological variability is acknowledged in the AFP process. How can the model be further advanced to not only simulate but quantitatively predict and control the extent of morphological variability, taking into account the inherent uncertainties in manufacturing processes?

4.     In the context of real-world manufacturing, how feasible is it to adapt the proposed framework for real-time monitoring and adaptive control of the AFP process to compensate for unexpected variations in material properties or manufacturing conditions without compromising the precision of the layup design?

5.     With the continuous development of advanced composite materials, how adaptable is the proposed framework to incorporate emerging materials with unique rheological properties or compositional complexities that may go beyond the current understanding of chemo-rheological interactions?

6.     Given the complexity of the manufacturing phenomena and the interconnected nature of various factors, how might machine learning algorithms be integrated into the modeling framework to autonomously learn and adapt to evolving manufacturing conditions, potentially improving the accuracy of predictions over time?

7.     The proposed methodology focuses on a detailed analysis at a certain scale. How might the challenges associated with integrating multi-scale modeling be addressed to bridge the gap between the microscopic and macroscopic behaviors of the material, especially in scenarios with varying fiber tow sizes or heterogeneous material distributions?

8.     Validation is crucial, but how could the proposed methodology be validated in environments that replicate real-world manufacturing conditions more accurately, accounting for factors such as temperature variations, ambient pressures, and the presence of contaminants, which might be challenging to simulate in a controlled laboratory setting?

Author Response

  1. Reviewer Question: Given the complexity of large nonlinear deformation in automated fiber placement (AFP), what novel computational techniques or algorithms beyond finite element methods could be explored to address the computational challenges associated with simulating and predicting such deformations accurately?

Author’s Response: Thank you for the question, it is an interesting subject to think about. To respond to the reviewer’s question, a smoothed finite element methodology and other meshfree methods can be investigated for use instead of a FEA model. S-FEM models has shown to be capable of modelling structural-fluid interaction. This can be used to simulate the flow of resin between the fibers which is not explicitly modeled in the current FEA model. The resulting dynamic fiber volume fraction can lead to a more accurate deformation of the composite laminates. This would require development of specialized software instead of the commercially available FEA software used for the model, because of this, the most optimal model to develop is using finite element analysis. Currently this topic is beyond the scope of the work, but it is a potential that can be investigated further. A sentence added to the paper presents an alternative to FEA in the following sentence:

“With a large number of factors involved in the model, future algorithms, such as smoothed-FEA, and machine learning, can be used to predict the deformation response of composites during manufacturing. However, the present methodology is a more accessible approach for the process modeling of composites”

  1. Reviewer Question: Considering the interconnected nature of morphological features like non-uniform ply thickness, fiber waviness, and variable fiber volume fraction, how can the model be extended to dynamically simulate the evolution of these features over multiple manufacturing cycles or in scenarios with rapidly changing manufacturing conditions?

Author’s Response: it is an interesting question and to reply to the reviewer’s question, More physics can be utilized to further enhance the interconnected finite element model. This includes explicitly modeling the mass transfer and porosity of the tow which would dynamically change the fiber volume fraction. The outflow of resin into gaps would simulate the development of the resin rich region. Furthermore, the temperature in the autoclave may be modeled, and a convective boundary condition can be used to dynamically simulate the heat transfer. The potential of the model has been acknowledged by the statement:

“The models presented in the present work can be decoupled, and can modified for a more accurate solution in the future. This includes a more accurate prediction of the mass transfer through consideration of porosity of the fibers and taking uncertainty into consideration with the stochastic initial parameters, such as fiber volume fraction.”

  1. Reviewer Question: Morphological variability is acknowledged in the AFP process. How can the model be further advanced to not only simulate but quantitatively predict and control the extent of morphological variability, taking into account the inherent uncertainties in manufacturing processes?

Author’s Response: It is a valid point, but morphological variability is difficult to fully control and predict and is currently beyond our work. However, stochasticity can be theoretically added to the model through the initial parameters, such as fiber volume fraction and fiber misalignment, and the properties of the composite, such as elasticity properties of the composite. Furthermore, a model for the transient flow of the resin could add some uncertainty in the prediction of the deformation and the strength. The variation between runs can result in a strength of the cured composite as a probability. The same sentence as the previous response has been used to explain how uncertainty can be taken into account in future models: “The models presented in the present work can be decoupled and can modified for a more accurate solution in the future. This includes a more accurate prediction of the mass transfer through consideration of porosity of the fibers and taking uncertainty into consideration with the stochastic initial parameters, such as fiber volume fraction.”

  1. Reviewer Question: In the context of real-world manufacturing, how feasible is it to adapt the proposed framework for real-time monitoring and adaptive control of the AFP process to compensate for unexpected variations in material properties or manufacturing conditions without compromising the precision of the layup design?

Author’s Response: The present process model will be used to predict the transient deformation which is difficult to measure in a non-destructive way during the manufacturing. The resulting morphology from the process model can be used to predict the strength of the composite part. The geometry after the process model is used to predict the strength through progressive failure analysis of the composite that includes the morphological features, such as resin rich regions and waviness. The prediction can then be used to optimize the layup for a better margin of safety. Real time monitoring is beyond the scope of the present work, since more physical phenomena would need to be represented for a more robust model. 

  1. Reviewer Question: With the continuous development of advanced composite materials, how adaptable is the proposed framework to incorporate emerging materials with unique rheological properties or compositional complexities that may go beyond the current understanding of chemo-rheological interactions?

Author’s Response: The components of the model can be decoupled. The chemo-rheological model and the heat transfer model can be reconfigured for any unique properties of the resin. Examples includes changing the temperature profile on the composite, the function used for the cure of the resin and the temperature dependent elastic components to predict the morphology. Furthermore, the iterative modeling outlined in the paper may be calibrated to a different material system and a stacking sequence. To ensure the internal resin pressure accurately predicts the size of the resin rich region. A sentence has been added to the conclusion to further affirm this answer:

“Moreover, the proposed multi-physics modeling framework can be adopted to accommodate various material systems with different chemo-rheological properties.”

  1. Reviewer Question: Given the complexity of the manufacturing phenomena and the interconnected nature of various factors, how might machine learning algorithms be integrated into the modeling framework to autonomously learn and adapt to evolving manufacturing conditions, potentially improving the accuracy of predictions over time?

Author’s Response: If a large amount of experimental data is available, a machine learning can be used to predict the deformation of material system. This would require understanding the factors factors (such as pressure, temperature, and macroscopic features such as fiber volume fraction) and the resulting deformation. The response of the deformation and would be predicted using the algorithm. This could also be utilized to predict the heat distribution through the structure based on the same factors.  However, the experimental measurement is difficult without destroying the manufactured composite and would be costly to create the experimental data set. However, currently testing is expensive and time consuming, so we opted to predict the morphology using FEA with data from smaller experimental samples. Currently, there is work being done with using machine learning on composites as referenced in the newly added sentence:

“With a large number of factors involved in the model, future algorithms, such as smoothed-FEA, and machine learning, can be used to predict the deformation response of composites during manufacturing. However, the present methodology is a more accessible approach for the process modeling of composites”

  1. Reviewer Question: The proposed methodology focuses on a detailed analysis at a certain scale. How might the challenges associated with integrating multi-scale modeling be addressed to bridge the gap between the microscopic and macroscopic behaviors of the material, especially in scenarios with varying fiber tow sizes or heterogeneous material distributions?

Author’s Response: A method to bridge the macroscopic and microscopic domains for the composite is using representative volume elements. Each of the elements of an FEA model can be modeled as an independent section and explicitly model the fibers and the resin. This would create a model with a higher resolution. However, the resulting computational requirements would be greater than the proposed FEA model. A sentence has been added to the section of the analytical fiber volume fraction acknowledging the alternative: “Modeling the flow of resin explicitly can be achieved with a higher resolution model. However, this would require computationally expensive representative volume elements, to model the microscopic behavior. The present analytical equation allows an adequate estimate to the evolution of the fiber volume fraction.”

  1. Reviewer Question: Validation is crucial, but how could the proposed methodology be validated in environments that replicate real-world manufacturing conditions more accurately, accounting for factors such as temperature variations, ambient pressures, and the presence of contaminants, which might be challenging to simulate in a controlled laboratory setting?

Author’s Response: Since observing the composite in a non-destructive way is difficult, only manufactured laminate can be tested. Defects are explicitly placed in the laminate and then fully cured according to the manufacturer’s cure cycle. A pristine component can then be manufactured to test the strength knockdown based on the embedded defect. The resulting morphology from the process model can be used to test the strength of the laminate in separate FEA model. This strength can be compared with the experimentally manufactured composites to validate the response of the model. This work is in progress and validation will be done alongside future models that will test the composite’s strength, such as progressive damage analysis as stated with the added sentence:

“The resulting morphology can be utilized in further analysis of the composite, such as through progressive failure analysis, to compare with fabricated laminates in the future.”